# The relationship between happiness and self-rated health: A population-based study of 19499 Iranian adults

Samira Mohammadi[1], Mahmoud Tavousi[1], Ali Asghar Haeri-Mehrizi[1], Fatemeh Naghizadeh Moghari[1], Ali Montazeri[1,2]*

1 Metrics Research Center, Iranian Institute for Health Sciences Research, ACECR, Tehran, Iran, 2 Faculty of Humanity Sciences, University of Science and Culture, Tehran, Iran

* montazeri@acecr.ac.ir

## Abstract

### Background

Health is one of the most important factors that affect happiness. This study aimed to assess the association between happiness and self-rated health among the Iranian adult population.

### Methods

This cross-sectional study was conducted on a sample of adults aged 18–65 In Iran in 2020. Demographic information, the Oxford Happiness Questionnaire, and a single item on self-rated health were used to collect the data. The relationship between happiness with demographic variables and self-rated health was examined by performing logistic regression analyses.

### Results

In all, 19499 were studied (9845 males and 9654 females). The mean age of participants was 36.38± 8.17 years. The mean happiness score was 4.1± 0.57 (out of 6) and this for self-rated health was 3.66±1.2 (out of 5). The results obtained from logistic regression analysis showed that very poor health status (OR: 5.114, 95% CI, P = 4.490–5.824, p <0.001), poor or very poor income status (OR: 1.553, 95% CI, P = 1.406–1.716, p <0.001), unemployment (OR: 1.704, 95% CI, P = 1.432–2.029, p <0.001), being aged 25–34 years (OR: 1.190, 95% CI, P = 1.088–1.302, p <0.001), and years of education (OR for 10–12 years of education: 1.271, 95% CI = 1.174–1.377, p< 0.001) were significant contributing factors to a lower happiness.

### Conclusion

The results showed that self-rated health was the most significant factor that affected happiness even after adjustment for socioeconomic variables, including age, income,

**Data Availability Statement:** All relevant data are within the manuscript and its Supporting information files.

**Funding:** AM was supported by the Elite Researcher Grant Committee under award number [982978] from the National Institutes for Medical Research Development (NIMAD), (www.nimad.ac.ir), Tehran, Iran. The funders had no role in study design, data collection and analysis, decision to publish, or preparation of the manuscript.

**Competing interests:** The authors declare that they have no competing interests.

employment, and education. Indeed, improving population health might be an effective measure to improve happiness among Iranians.

## Introduction

The World Health Organization emphasizes that happiness is an essential factor in the concept of health [1,2] and is one of the main components of life satisfaction [3]. People who consider themselves happy have better physical health than people who think themselves unhappy [4]. Veenhoven defines happiness as 'the degree to which an individual judges the overall quality of his or her life-as-a-whole favorably'; in other words, 'how much one likes the life one lives' [5].

According to the world happiness report (2017–2019), the highest and lowest happiness scores were for Finland and Afghanistan, respectively. The Islamic Republic of Iran ranked 118th among 153countries. Although happiness score in Iran was lower than some countries in the Eastern Mediterranean Region (EMRO), such as Saudi Arabia, Pakistan, Morocco, but was higher than some other countries including Jordan, Tunisia, and Egypt [6].

One of the most frequently used measures of self-reported health status is a single question asking individuals to rate their overall health on a scale from excellent to very poor. There is widespread agreement that this simple global question provides a useful summary of how individuals perceive their overall health status [7]. The results of a cross-national study that compared health in Egypt, Iran, Jordan, and the United States showed that means and standard deviations of self-rated health by country was (2.79±0.85), (2.99±0.81), (3.06±0.83), and (3.23±0.78), respectively [8].

The association between happiness and health or association between happiness and some health-related behaviors are well documented. For instance, a review on subjective well-being (popularly referred to as happiness or life satisfaction) reported that higher subjective well-being was associated with good health and longevity, better social relationships, work performance, and creativity [9]. Also, a recent study from 15 European countries reported that compared to inactive people, there was a positive dose-response association between physical activity volume and happiness [10]. A review including longitudinal and experimental studies, have found strong associations between happiness and health outcomes [11] such as death, coronary heart disease (CHD) [12,13], stroke [14], type 2 diabetes [15] and life expectancy [16]. As such, it is argued that impaired happiness might be a consequence of ill-health but also could potentially contributor to the risk of several diseases since happiness includes affective well-being (feelings of joy and pleasure), eudemonic well-being (sense of meaning and purpose in life), and evaluative well-being (life satisfaction) [17].

Although an association between health status and happiness depends on how health is measured [18], a study showed that the relationship between happiness and self-rated health is somewhat more robust than the correlations between happiness and medical examinations [19]. Self-rated health is a widespread health measure that is based on personal perception of one's about their own health. It is a reliable measure, especially when objective data are insufficient to reflect disease severity or in patients with the undiagnosed disease [20].

Although there are several many Iranian studies on happiness [21], it seems that the relationship between happiness and health is overlooked. There is no national study in Iran, unlike other countries [22–26]. Thus to fill the gap, this national study aimed to investigate the

relationship between happiness and health in Iran to contribute public health, health policy, and the existing knowledge on the topic.

## Materials and methods

### Design and participants

This was a national cross-sectional study conducted from 10 January to 20 January 2020 throughout all provinces in Iran. Currently Iran has 32 provinces with over 80 million populations. The inclusion criteria were aged 18–65, Iranian nationality, able to respond to the study questionnaires. No other restrictions were implemented.

### Sample size and sampling

The following formula was used to estimate the sample size:

$$n = \frac{z_{\alpha/2}^2 \ pq}{(rq)^2}$$

To estimate the sample size, according to a national study [27], and based on population density, the country was classified into five categories. Then, samples were selected based on multi-stage sampling from each category. In doing so, one province was randomly selected from each category. Then two cities and two rural settings were randomly selected in each province. Every household within the city and rural areas had the same probability of being sampled. The households to be sampled were selected using systematic sampling within each census section. Finally, sampling units (the individuals) were selected randomly from all eligible persons living in the same household. Informed consent was obtained from each individual after the purpose of the study was explained. Considering the effect size of 1.4, the sample size of 20320 was estimated. However, in practice, 19499 Iranian adults were entered into the study.

### Measures

1. The Oxford standard Happiness Questionnaire (OHQ): The overall happiness score could be derived from some of the items scores divided by 29, giving a score range from 1 to 6 (strongly disagree = 1, moderately disagree = 2, slightly disagree = 3, slightly agree = 4, moderately agree = 5, strongly agree = 6). The scores are interpreted as follows: Unhappy (1 to <2); moderately unhappy (2 to < 3); not very happy/unhappy (3 to <4); somewhat happy/moderately happy (4); rather happy/pretty happy (4 to <5); very happy (5 to <6); too happy (6) [28]. Psychometric properties of the Iranian version of questionnaire are well documented. Cronbach's alpha coefficient (measure of internal consistency) and interclass correlation coefficient (measure of stability) were 0.90 and 0.79, respectively. The convergent and divergent validity of the questionnaire were high and acceptable [29].

   2. Self-rated health: Self-rated health was assessed by a single item asking people to rate their general health status at present. Respondents self-rated their health status on a 5-point Likert scale as follows: very poor = 1, poor = 2, fair = 3, good = 4, very good = 5. The validity of self-rated health measures has been proven in several studies [30–32]. Validity and reliability of self-rated health measure among Iranian showed acceptable results. The criterion validity showed that the self-rated health and the WHO-5 well-being had positive correlation as expected (r = 0.5, p< 0.001). Additionally, the reliability of the self-rated health, using interclass correlation coefficient (ICC), was found to be 0.83; 95% CI (0.72 to 0.90) [32].

## Statistical analysis

Data were explored using descriptive statistics, including frequency, percentage, mean and standard deviation. The missing data were replaced with each item series mean. Logistic regression analyses were performed to assess the relationship between happiness and independent variables, including participants' health status. However, since some eminent scholars [33] believe that there is an auto-correlation between item 28 and the self-rated health, we did calculate, and reanalyze the data while item 28 (I do not feel particularly healthy) was excluded from the Oxford happiness score. As such for both with and without item 28 of the Oxford questionnaire, happiness as dependent variables were categorized into: 'happy' (scores ranging from 4 to 6) and 'unhappy' (scores ranging from 1 to 3). The results expressed as odds ratio and 95% confidence intervals. A significant level was set at $P < 0.05$. Since this questionnaire covers broader matters than happiness [33], we calculated correlations for the few items on life satisfaction with self-rated health separately.

## Ethics statement

The National Institutes for Medical Research Development (NIMAD), Tehran, Iran. ethics committee approved the study (IR.NIMAD.REC.l398.228). Due to the study design and all participants gave their verbal consent.

## Results

A total of 19499 Iranian adults participated in the study (9845 males and 9654 females). The mean age of participants was 36.38 ± 8.17 years and the mean years of education were 10.51 ±4.43. Demographic details of the participants are presented in Table 1.

The results showed that the mean happiness score was 4.1 ± 0.57 (out of 6), and this was 3.66±1.2 (out of 5) for self-rated health. Overall, 51.6% of the respondents scored more than four and less than 5, indicating a rather happy/pretty happy condition (4> to <5). The findings also showed that 34.8% of the participants reported their health status as very good, and % 6.9 reported their health status as very poor. The detailed results are shown in Table 2. In addition, the results obtained from correlation between few items on life satisfaction and self-rated health in are presented in Table 3. Overall the findings showed a relatively low correlation (r = 0.15).

The results obtained from logistic regression analysis showed that people with very poor health status (OR: 5.114, 95% CI, P = 4.490–5.824, p <0.001), people with poor or very poor income level (OR: 1.553, 95% CI, P = 1.406–1.716, p <0.001), unemployed (OR: 1.704, 95% CI, P = 1.432–2.029, p <0.001), aged (OR: 1.190, 95% CI, P = 1.088–1.302, p <0.001), and people with 10–12 years of education (OR:1.271, 95% CI, P = 1.174–1.377, p <0.001) were more likely to report a lower score for happiness. The results are shown in Table 4. In addition, the results obtained from the same analysis when items 28 was excluded are shown in Table 5. The results almost were very similar and no significant difference was observed from the previous analysis except age for 18–24 (OR:1.191, 95% CI, P = 1.038–1.367, p = 0.013) and 6–9 years of education (OR:1.118, 95% CI, P = 1.024–1.221, p = 0.013).

## Discussion

This study investigated the relationship between happiness and self-rated health in 19499 adults aged 18 to 65 years in Iran. To the best of our knowledge, this is the first Iranian national study that investigates the relationship between happiness and self-rated health among adult populations in Iran, a country located in a conflict area, and faces several challenges, including

**Table 1. Frequency distributions of the participants' demographic characteristics.**

|  | Female (n = 9845) | Male) n = 9654 | Total) n = 19499 |
|---|---|---|---|
|  | No. (%) | No. (%) | No. (%) |
| **Age(year)** |  |  |  |
| 18–24 | 982(10.0) | 278(2.9) | 1260(6.5) |
| 25–34 | 3914(39.8) | 3190(33.0) | 7104(36.4) |
| 35–44 | 3614(36.7) | 3846(39.8) | 7460(38.3) |
| 45–65 | 1335(13.6) | 2340(24.2) | 3675(18.8) |
| **Education (year)** |  |  |  |
| 1–5 | 2062(20.9) | 1353(14.0) | 3415(17.5) |
| 6–9 | 2163(22.0) | 2211(22.9) | 4374(22.4) |
| 10–12 | 3304(33.6) | 3109(32.2) | 6413(32.9) |
| 13≤ | 2316(23.5) | 2981(30.9) | 5297(27.2) |
| **Employment status** |  |  |  |
| Employed | 1150(11.7) | 8478(87.8) | 9628(49.4) |
| Housewife | 8470(86.0) | 0(0.0) | 8470(43.4) |
| Retired | 39(0.4) | 389(4.0) | 428 (2.2) |
| Student | 180(1.8) | 151(1.6) | 331(1.7) |
| Unemployed | 6(0.1) | 636(6.6) | 642(3.3) |
| **Income (self-reported)** |  |  |  |
| Very good/ good | 2323(23.6) | 2007(20.8) | 4330(22.2) |
| Intermediate | 5962(60.6) | 5862(60.7) | 11824(60.6) |
| Very poor/poor | 1560(15.8) | 1785(18.5) | 3345(17.2) |

**Table 2. Overall distribution of Oxford happiness scores and self-rated health scores (n = 19499).**

|  | Frequency | Percent |
|---|---|---|
| **Happiness (score range)** |  |  |
| Unhappy (1 to < 2) | 11 | 0.1 |
| Moderately unhappy (2 to < 3) | 474 | 2.4 |
| Not very happy/unhappy (3 to <4) | 7600 | 39.0 |
| Somewhat happy/moderately happy (4) | 329 | 1.7 |
| Rather happy/pretty happy (4> to <5) | 10059 | 51.6 |
| Very happy (5 to <6) | 1022 | 5.2 |
| Too happy (6) | 4 | 0.0 |
| Mean (SD) | 4.10 (0.57) |  |
| **Happiness without item 28** |  |  |
| Mean (SD) | 4.09 (0.57) |  |
| **Self-rated health** |  |  |
| Very poor | 1348 | 6.9 |
| Poor | 2349 | 12.0 |
| Fair | 4494 | 23.0 |
| Good | 4530 | 23.2 |
| Very good | 6778 | 34.8 |
| Mean (SD) | 3.66 (1.2) | - |

**Table 3. Correlation between the oxford happiness questionnaire few items on life satisfaction and self-rated health.**

| | Correlation coefficient |
|---|---|
| I don't feel particularly pleased with the way I am(R) | 0.08 |
| I feel that life is very rewarding | 0.12 |
| Life is good | 0.13 |
| I do not think that the world is a good place (R) | 0.11 |
| I am well satisfied about everything in my life | 0.11 |
| Total | 0.15 |

economic sanctions. However, the data presented in the current study confirmed that happiness, to a large extent, is dependent on health and some socioeconomic factors related to income and employment. Specifically, although the findings add a little to knowledge, the main rationale for the study could be that this relation was explored in the Iranian context. In doing so it was suggested to examine the relationship between item 15 of the Oxford

**Table 4. Relationship between Oxford happiness scores, health status and demographic variables.**

| | Univariate regression | | Multivariate p value regression | |
|---|---|---|---|---|
| | OR(95% CI) | p value | OR(95% CI) | p value |
| **Age (years)** | | | | |
| 45–65 | 1.00 (ref) | - | 1.00(ref) | - |
| 35–44 | 1.203 (1.109–1.305) | <0.001 | 1.161(1.064–1.268) | 0.001 |
| 25–34 | 1.251 (1.152–1.357) | <0.001 | 1.190(1.088–1.302) | <0.001 |
| 18–24 | 1.274(1.118–1.405) | <0.001 | 1.136(0.987–1.307) | 0.075 |
| **Gender** | | | | |
| Male | 1.00 (ref) | - | 1.00(ref) | - |
| Female | 1.117(1.055–1.183) | <0.001 | 1.021(0.899–1.160) | 0.747 |
| **Education level (years)** | | | | |
| 13≤ | 1.00(ref) | - | 1.00(ref) | - |
| 10–12 | 1.334(1.238–1.437) | <0.001 | 1.271(1.174–1.377) | <0.001 |
| 6–9 | 1.130(1.041–1.227) | 0.004 | 1.041(0.951–1.139) | 0.387 |
| 1–5 | 1.284(1.176–1.401) | <0.001 | 1.129(1.024–1.245) | 0.015 |
| **Employment** | | | | |
| Employed | 1.00(ref) | - | 1.00(ref) | - |
| Housewife | 1.231(1.160–1.306) | <0.001 | 1.231(1.079–1.403) | 0.002 |
| Retired | 0.798(0.650–0.979) | 0.031 | 0.877(0.703–1.093) | 0.243 |
| Student | 1.333(1.070–1.662) | 0.010 | 1.430(1.130–1.810) | 0.003 |
| Unemployed | 2.261(1.922–2.660) | <0.001 | 1.704(1.432–2.029) | <0.001 |
| **Income status(self-reported)** | | | | |
| Good/Very good | 1.00(ref) | - | 1.00(ref) | - |
| Intermediate | 1.030(0.959–1.106) | 0.418 | 0.997(0.925–1.075) | 0.937 |
| Poor/Very poor | 1.708(1.559–1.871) | <0.001 | 1.553(1.406–1.716) | <0.001 |
| **Self-rated health** | | | | |
| Very good | 1.00(ref) | - | 1.00(ref) | - |
| Good | 1.147(1.060–1.242) | 0.001 | 1.144(1.056–1.239) | 0.001 |
| Fair | 1.419(1.312–1.534) | <0.001 | 1.417(1.310–1.534) | <0.001 |
| Poor | 3.062(2.779–3.373) | <0.001 | 3.085(2.798–3.403) | <0.001 |
| Very Poor | 5.073(4.461–5.768) | <0.001 | 5.114(4.490–5.824) | <0.001 |

**Table 5. Relationship between Oxford happiness scores (without item 28), health status and demographic variables.**

| | Univariate regression | | Multivariate p value regression | |
|---|---|---|---|---|
| | OR(95% CI) | p value | OR(95% CI) | p value |
| **Age (years)** | | | | |
| 45–65 | 1.00 (ref) | - | 1.00(ref) | - |
| 35–44 | 1.230 (1.136–1.332) | <0.001 | 1.200(1.102–1.307) | <0.001 |
| 25–34 | 1.254 (1.158–1.359) | <0.001 | 1.211(1.110–1.323) | <0.001 |
| 18–24 | 1.315(1.157–1.495) | <0.001 | 1.191(1.038–1.367) | 0.013 |
| **Gender** | | | | |
| Male | 1.00 (ref) | - | 1.00(ref) | - |
| Female | 1.123(1.061–1.188) | <0.001 | 1.033(0.913–1.169) | 0.609 |
| **Education level (years)** | | | | |
| 13≤ | 1.00(ref) | - | 1.00(ref) | - |
| 10–12 | 1.374(1.277–1.479) | <0.001 | 1.308(1.209–1.414) | <0.001 |
| 6–9 | 1.215(1.121–1.316) | <0.001 | 1.118(1.024–1.221) | 0.013 |
| 1–5 | 1.326 (1.216–1.445) | <0.001 | 1.168(1.062–1.285) | 0.001 |
| **Employment** | | | | |
| Employed | 1.00(ref) | - | 1.00(ref) | - |
| Housewife | 1.234(1.164–1.309) | <0.001 | 1.192(1.049–1.354) | 0.007 |
| Retired | 0.815(0.669–0.993) | 0.042 | 0.916(0.742–1.131) | 0.417 |
| Student | 1.332(1.069–1.659) | 0.010 | 1.415(1.121–1.787) | 0.004 |
| Unemployed | 2.212(1.872–2.614) | <0.001 | 1.623(1.360–1.938) | <0.001 |
| **Income status(self-reported)** | | | | |
| Good/Very good | 1.00(ref) | - | 1.00(ref) | - |
| Intermediate | 1.069(0.996–1.146) | 0.063 | 1.032(0.959–1.110) | 0.400 |
| Poor/Very poor | 1.812(1.654–1.986) | <0.001 | 1.642(1.488–1.812) | <0.001 |
| **Self-rated health** | | | | |
| Very good | 1.00(ref) | - | 1.00(ref) | - |
| Good | 1.125(1.043–1.214) | 0.002 | 1.123(1.040–1.213) | 0.003 |
| Fair | 1.326(2.229–1.431) | <0.001 | 1.324(1.226–1.429) | <0.001 |
| Poor | 2.653(2.407–2.925) | <0.001 | 2.672(2.421–2.984) | <0.001 |
| Very Poor | 4.134(3.626–4.714) | <0.001 | 4.148(3.633–4.736) | <0.001 |

Happiness Questionnaire (I am very happy) and self-rated health and see how the correlation compares with similar findings among other nations. The result showed that the correlation between item 15 and self-rated health is about 0.12 well below findings from other countries (see S1 Appendix) [34].

Although 2020 coincided with the Covid-19 pandemic, and Covid-19 as a health-threatening factor can affect the level of happiness [35], we were fortunate to collect the data before the pandemic began in Iran. The first tow of deaths related to COVID-19 was reported on February 19, 2020, in Iran [36] while we collected the data in early January 2020.

The findings suggest that individuals who reported lower health status were more likely to report unhappiness even after controlling for various demographic and socioeconomic factors. This result is in good agreement with those of previous investigations where it has been reported that self-rated health is an important determinant of happiness, or quantitatively much more important than other demographic and economic characteristics [37,38]. Similar studies, showing that when people suffer from a severe illness or are in pain, their capacity for

happiness is impaired. For example, chronic physical disease, mood, anxiety, and other mental disorders can significantly reduce happiness [39].

Also, the current study showed that employment status and income independently influenced happiness. Unemployed people are more likely to be unhappy than employed people, and people who reported lower income levels were more likely to be unhappy than those who had higher income. These results support a previous study that demonstrated that happy people were more likely to be employed and earn more than unhappy people [40]. The finding from another study that investigated the association between income and happiness in East and South Asia, including China, Singapore, Japan, India, Malaysia, Philippines, South Korea, Thailand, Taiwan, Hong Kong, showed that associations between income and happiness were strongly significant in some countries, including South Korea and Taiwan [41]. Thus policies aimed at increasing employment and reducing income disparities may increase happiness in individuals.

## Strengths and limitations

The study benefited from relatively large sample size and included people living in rural and urban communities with different socio-economic backgrounds. Another notable strength was the precision of the data collection by the trained interviewers. However, one might consider the probability of a socially desirable response behavior in the direct contact between interviewers and respondents. Thus the interpretation of the results needs caution.

Multistage sampling can simplify data collection when we have large, geographically spread samples, and we can obtain a probability sample without a complete sampling frame. However, it can lead to unrepresentative samples because large sections of populations may not be selected for sampling. Since we used multistage sampling, the result might not be generalized to all Iranians.

One should bear in mind that the Oxford Happiness Questionnaire covers a wide range of traits rather than happiness in the sense of life satisfaction [33]. Perhaps in future studies, if we are going to measure happiness in the sense of life satisfaction, there is a need to use an appropriate measure. In addition, item 28 of the Oxford Happiness Questionnaire is about self-rated health, and thus it might cause autocorrelation with the self-rated health measure, although the item 28 and the measure of self-rated health are worded differently. The latter is negative (I do not feel particularly healthy) while the former is positive and askes people to rate their current health. However, as indicated in reanalysis of the data (Table 5), the findings did not show any major differences to our earlier analysis as shown in Table 4.

## Conclusion

The results obtained from the current study confirmed that a strong relationship exists between health and happiness. Self-rated health was the most influencing factor affecting happiness even after adjusting for socioeconomic variables. It seems that adopting policies to improve public health and placing health on the public agenda could be an effective approach for increasing happiness.

## Supporting information

**S1 Appendix. The correlations between health and happiness in selected studies.**
(DOCX)

**S1 Dataset.**
(SAV)

## Acknowledgments

The authors are grateful to all participants, who made this study possible.

## Author Contributions

**Conceptualization:** Samira Mohammadi, Mahmoud Tavousi, Ali Montazeri.

**Data curation:** Samira Mohammadi, Ali Asghar Haeri-Mehrizi, Fatemeh Naghizadeh Moghari, Ali Montazeri.

**Formal analysis:** Ali Asghar Haeri-Mehrizi, Ali Montazeri.

**Funding acquisition:** Ali Montazeri.

**Investigation:** Samira Mohammadi, Ali Montazeri.

**Methodology:** Samira Mohammadi, Mahmoud Tavousi, Ali Montazeri.

**Project administration:** Mahmoud Tavousi, Ali Montazeri.

**Resources:** Ali Montazeri.

**Software:** Ali Montazeri.

**Supervision:** Ali Montazeri.

**Validation:** Samira Mohammadi, Ali Montazeri.

**Visualization:** Samira Mohammadi, Ali Montazeri.

**Writing – original draft:** Samira Mohammadi, Ali Montazeri.

**Writing – review & editing:** Ali Montazeri.

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
