## [Decision Letter · Decision Letter 0]

1 Dec 2021

PONE-D-21-28962The relationship between happiness and self-rated health: A population-based study of 19499 Iranian adultsPLOS ONE

Dear Dr. Montazeri,

Thank you for submitting your manuscript to PLOS ONE. After careful consideration, we feel that it has merit but does not fully meet PLOS ONE’s publication criteria as it currently stands. Therefore, we invite you to submit a revised version of the manuscript that addresses the points raised during the review process. According to the reviewers’ comments and my evaluation, the methods and discussion sections need careful attention and must be improved according to the comments. 

We look forward to receiving your revised manuscript.

Kind regards,

Forough Mortazavi

Academic Editor

PLOS ONE

Journal Requirements:

Additional Editor Comments (if provided):

Dear authors,

Thank you for submitting your manuscript to PLOS ONE. According to the reviewers’ comments and my evaluation, several points needs careful attention.

1. The abstract should be rewritten using a structured format.

2. In the methods section, the authors should describe the sampling method and the procedures for selecting the participants in detail.

3. This study was performed during the covid-19 pandemic. This may have effected on the selection of the sample.

4. The pandemic may have affected both the level of happiness and perceived health. This should be taken into consideration in comparisons with the findings of previous studies in the discussion section.

5. PLS consider the STROBE checklist for reports of observational studies and revise the manuscript taking the following points into account:

6. Describe the setting, locations, and relevant dates, including periods of data collection.

7. Describe any efforts to address potential sources of bias.

8. Explain how missing data were addressed.

9. Discuss limitations of the study, taking into account sources of potential bias or imprecision.

10. Discuss both direction and magnitude of any potential bias.

11. Discuss the generalizability of the study results.

12. Limitations must be further explained in relation to sampling method.

Reviewers' comments:

Reviewer's Responses to Questions

**Comments to the Author**

1. Is the manuscript technically sound, and do the data support the conclusions?

Reviewer #1: Partly

Reviewer #2: Yes

2. Has the statistical analysis been performed appropriately and rigorously? 

Reviewer #1: Yes

Reviewer #2: Yes

3. Have the authors made all data underlying the findings in their manuscript fully available?

Reviewer #1: No

Reviewer #2: Yes

4. Is the manuscript presented in an intelligible fashion and written in standard English?

Reviewer #1: Yes

Reviewer #2: No

5. Review Comments to the Author

Reviewer #1: This is an elementary report of a well-known relationship and is OK in itself. but adds little to knowledge.

The main rationale is to show this relation in Iranian context. In the context of that goal the authors should compare the strenghts of the correlations with similar studies conducted in other countries.

Doing so, they might consider to compare with correlations between happiness and health gathered in the World Database of Happiness, at https://worlddatabaseofhappiness.eur.nl/search-the-database/correlational-findings/#id=-1yD-HsBSlHDfFpgD2EY. This would require that they use item 15 from the OHQ rather than the full scale.

The use of the OHQ should be reconsidered anyway. A look at the items shows that it covers a wide range of positive traits rather than happiness in the sense of life-satisfaction. Item 28 is about self-rated health and causes autocorrelation.

In the discussion section the authos acknowledge that the correlation may be driven by an effect of happiness on health, but in the conclusion they attribute the correlation to the effects of health on happiness

Reviewer #2: The results of this paper are interesting, and can contribute to the literature on happiness and health among adults.

However, several revisions are required for this work to be accurately portrayed, received, and interpreted. Below, I have separated my comments into an overall comment.

- The first paragraph should include some comparative statistics on figures/statistics from EMRO region countries to provide the context for Iran being similar (in terms of happiness and health).

- The validity and reliability of the questionnaire (Oxford standard Happiness Questionnaire & Self-rated health) among the Iranian population should be justified in detail.

- In table 1; write the scale of Education & Income.

- More discussion about the justification of finding should be provided in this section. The discussion is disorganized and confusing. It is better to summarize your findings firstly and then discuss them separately.

- The conclusion is the repetition of what has been said in the text! And need a revision based on results.

- The manuscript needs a revision for grammar, typos, and English expressions to improve the readability.

6. PLOS authors have the option to publish the peer review history of their article (what does this mean?). If published, this will include your full peer review and any attached files.

Reviewer #1: **Yes: **Ruut Veenhoven

Reviewer #2: **Yes: **Vahid Rashedi

---

## [Author Response · Author response to Decision Letter 0]

17 Jan 2022

Forough Mortazavi

Academic Editor

PLOS ONE

12 January 2022

Dear Dr. Mortazavi,

PONE-D-21-28962

The relationship between happiness and self-rated health: A population-based study of 19499 Iranian adults

PLOS ONE

Thank you for your e-mail. We are grateful to both reviewers and found their comments very helpful. We have revised the manuscript and now very pleased to submit the revised version for your consideration. All changed are marked in blue and hope you find it satisfactory.

Wish you all the best.

Yours sincerely

Ali Montazeri

Thank you. Done.

This following sentences add to the manuscript.

Data Availability 

The datasets generated for this study are fully available without restriction.

Done.

Ethics statement

The National Institutes for Medical Research Development (NIMAD), Tehran, Iran. ethics committee approved the study (IR.NIMAD.REC.l398.228). Due to the study design and all participants gave their verbal consent.

Additional Editor Comments

Dear authors,

Thank you for submitting your manuscript to PLOS ONE. According to the reviewers’ comments and my evaluation, several points needs careful attention.

1. The abstract should be rewritten using a structured format.

Done.

2. In the methods section, the authors should describe the sampling method and the procedures for selecting the participants in detail.

The following sentences were added to the Methods as requested:

To estimate the sample size, according to a national study [30], and based on population density, the country was classified into five categories. Then, samples were selected based on multi-stage sampling from each category. In doing so, one province was randomly selected from each category. Then two cities and two rural settings were randomly selected in each province. Every household within the city and rural areas had the same probability of being sampled. The households to be sampled were selected using systematic sampling within each census section. Finally, sampling units (the individuals) were selected randomly from all eligible persons living in the same household. Informed consent was obtained from each individual after the purpose of the study was explained. Considering the effect size of 1.4, the sample size of 20320 was estimated. However, in practice, 19499 Iranian adults were entered into the study.

3. This study was performed during the covid-19 pandemic. This may have effected on the selection of the sample.

The study was carried out just before pandemic. To satisfy the reviewer’s comment we added the following sentences to the first paragraph of Discussion:

Although 2020 coincided with the Covid-19 pandemic, and Covid-19 as a health-threatening factor can affect the level of happiness [37], we were fortunate to collect the data before the pandemic began in Iran. The first tow of deaths related to COVID-19 was reported on February 19, 2020, in Iran [38] while we collected the data in early January 2020.

4. The pandemic may have affected both the level of happiness and perceived health. This should be taken into consideration in comparisons with the findings of previous studies in the discussion section.

Please see the above. 

5. PLS consider the STROBE checklist for reports of observational studies and revise the manuscript taking the following points into account:

6. Describe the setting, locations, and relevant dates, including periods of data collection.

Done.

This was a national cross-sectional study conducted from 10 January to 20 January 2020 throughout all provinces in Iran. Currently Iran has 32 provinces with over 80 million populations. 

7. Describe any efforts to address potential sources of bias.

Done.

Please see strengths and limitations.

8. Explain how missing data were addressed.

Done.

Please see strengths and limitations.

9. Discuss limitations of the study, taking into account sources of potential bias or imprecision.

Done.

Please see strengths and limitations.

10. Discuss both direction and magnitude of any potential bias.

Done.

Please see strengths and limitations.

11. Discuss the generalizability of the study results.

Done.

Please see strengths and limitations.

12. Limitations must be further explained in relation to sampling method.

Thank you. There was no limitation in relation to sampling method. However, we included a separate section on strengths and limitations. Hope you find it satisfactory.

Please see strengths and limitations. Multistage sampling can simplify data collection when we have large, geographically spread samples, and we can obtain a probability sample without a complete sampling frame. However, it can lead to unrepresentative samples because large sections of populations may not be selected for sampling. Since we used multistage sampling, the result might not be generalized to all Iranians.

Review Comments to the Author

Reviewer #1 Prof. Ruut Veenhoven

This is an elementary report of a well-known relationship and is OK in itself. but adds little to knowledge. The main rationale is to show this relation in Iranian context. In the context of that goal the authors should compare the strengths of the correlations with similar studies conducted in other countries. Doing so, they might consider to compare with correlations between happiness and health gathered in the World Database of Happiness, at https://worlddatabaseofhappiness.eur.nl/search-the-database/correlational-findings/#id=-1yD-HsBSlHDfFpgD2EY. This would require that they use item 15 from the OHQ rather than the full scale.

Specifically, although the findings add a little to knowledge, the main rationale for the study could be that this relation was explored in the Iranian context. In doing so it was suggested to examine the relationship between item 15 of the Oxford Happiness Questionnaire (I am very happy) and self-rated health and see how the correlation compares with similar findings among other nations. The result showed that the correlation between item 15 and self-rated health is about 0.64 and is very similar with other studies (see S1 Appendix) [36].

The use of the OHQ should be reconsidered anyway. A look at the items shows that it covers a wide range of positive traits rather than happiness in the sense of life-satisfaction. Item 28 is about self-rated health and causes autocorrelation.

We agree. However, at this stage it is impossible to reconsider the happiness measurement. However, we added the following statement to the limitation to comply with the recommendation:

One should bear in mind that the Oxford Happiness Questionnaire covers a wide range of traits rather than happiness in the sense of life satisfaction [44]. Perhaps in future studies, if we are going to measure happiness in the sense of life satisfaction, there is a need to use an appropriate measure. In addition, item 28 of the Oxford Happiness Questionnaire is about self-rated health, and thus it might cause autocorrelation with the self-rated health measure, although the item 28 and the measure of self-rated health are worded differently. The latter is negative (I do not feel particularly healthy) while the former is positive and askes people to rate their current health.

In the discussion section the authors acknowledge that the correlation may be driven by an effect of happiness on health, but in the conclusion they attribute the correlation to the effects of health on happiness.

Thank you for your comment. We agree that we made a mistake and thus the paragraph was deleted.

Reviewer #2: Dr. Vahid Rashedi

The results of this paper are interesting, and can contribute to the literature on happiness and health among adults. However, several revisions are required for this work to be accurately portrayed, received, and interpreted. Below, I have separated my comments into an overall comment.

Thank you for your positive evaluation and comments.

- The first paragraph should include some comparative statistics on figures/statistics from EMRO region countries to provide the context for Iran being similar (in terms of happiness and health).

According to the world happiness report (2017-2019), the highest and lowest happiness scores were for Finland and Afghanistan, respectively. The Islamic Republic of Iran ranked 118th among 153countries. Although happiness score in Iran was lower than some countries in the Eastern Mediterranean Region (EMRO), such as Saudi Arabia, Pakistan, Morocco, but was higher than some other countries including Jordan, Tunisia, and Egypt [6]. 

One of the most frequently used measures of self-reported health status is a single question asking individuals to rate their overall health on a scale from excellent to very poor. There is widespread agreement that this simple global question provides a useful summary of how individuals perceive their overall health status [7]. The results of a cross-national study that compared health in Egypt, Iran, Jordan, and the United States showed that means and standard deviations of self-rated health by country was (2.79±0.85), (2.99±0.81), (3.06±0.83), and (3.23±0.78), respectively [8].

- The validity and reliability of the questionnaire (Oxford standard Happiness Questionnaire & Self-rated health) among the Iranian population should be justified in detail.

Thank you. Both of them were added in detail.

Psychometric properties of the Iranian version of questionnaire are well documented. Cronbach's alpha coefficient (measure of internal consistency) and interclass correlation coefficient (measure of stability) were 0.90 and 0.79, respectively. The convergent and divergent validity of the questionnaire were high and acceptable [32].

Validity and reliability of self-rated health measure among Iranian showed acceptable results. The criterion validity showed that the self-rated health and the WHO-5 well-being had positive correlation as expected (r= 0.5, p< 0.001). Additionally, the reliability of the self-rated health, using interclass correlation coefficient (ICC), was found to be 0.83; 95% CI (0.72 to 0.90) [35].

- In table 1; write the scale of Education & Income.

The scales were included.

Education: year

Income: self-reported

- More discussion about the justification of finding should be provided in this section. The discussion is disorganized and confusing. It is better to summarize your findings firstly and then discuss them separately.

Done.

Please see discussion

- The conclusion is the repetition of what has been said in the text! And need a revision based on results.

Thank you for your comment. The conclusion was revised.

It seems that adopting policies to improve public health and placing health on the public agenda could be an effective approach for increasing happiness.

- The manuscript needs a revision for grammar, typos, and English expressions to improve the readability.

Thank you. The manuscript was copy edited once more.

---

## [Decision Letter · Decision Letter 1]

27 Jan 2022

PONE-D-21-28962R1The relationship between happiness and self-rated health: A population-based study of 19499 Iranian adultsPLOS ONE

Dear Dr. Montazeri,

Thank you for submitting your manuscript to PLOS ONE. After careful consideration, we feel that it has merit but does not fully meet PLOS ONE’s publication criteria as it currently stands. Therefore, we invite you to submit a revised version of the manuscript that addresses the points raised during the review process.

We look forward to receiving your revised manuscript.

Kind regards,

Forough Mortazavi

Academic Editor

PLOS ONE

Additional Editor Comments:

Dear authors,

Thank you for revising the manuscript according to the reviewers’ comments; however, a few points still remain. Please kindly consider the points raised by reviewer 1 regarding item 28 and auto-correlation. In presenting the results of statistical analyses, please describe how the authors divided the sample into two groups of low and high happiness for logistic regression analyses. This section is in need of further clarification. Also, in the titles of tables 2 and 3, PLS replace 'happiness' with the correct description, i.e. 'Oxford happiness scores’.

Regards,

Reviewers' comments:

Reviewer's Responses to Questions

**Comments to the Author**

1. If the authors have adequately addressed your comments raised in a previous round of review and you feel that this manuscript is now acceptable for publication, you may indicate that here to bypass the “Comments to the Author” section, enter your conflict of interest statement in the “Confidential to Editor” section, and submit your "Accept" recommendation.

Reviewer #1: (No Response)

Reviewer #2: All comments have been addressed

2. Is the manuscript technically sound, and do the data support the conclusions?

Reviewer #1: Partly

Reviewer #2: Yes

3. Has the statistical analysis been performed appropriately and rigorously? 

Reviewer #1: Yes

Reviewer #2: Yes

4. Have the authors made all data underlying the findings in their manuscript fully available?

Reviewer #1: (No Response)

Reviewer #2: Yes

5. Is the manuscript presented in an intelligible fashion and written in standard English?

Reviewer #1: Yes

Reviewer #2: Yes

6. Review Comments to the Author

Reviewer #1: My objection about auto-correlation is not met. The answer that the concerned questions on health were formulated differently does not convince. Without re-analysis of the data I cannot recommend acceptance.

Reviewer #2: I read the manuscript with great interest and think the data reported in this study is valuable, and the authors have made all the corrections to the article entitled: "The relationship between happiness and self-rated health: A population-based study of 19499 Iranian adults".

7. PLOS authors have the option to publish the peer review history of their article (what does this mean?). If published, this will include your full peer review and any attached files.

Reviewer #1: **Yes: **Ruut Veenhoven

Reviewer #2: **Yes: **Vahid Rashedi

---

## [Author Response · Author response to Decision Letter 1]

12 Feb 2022

Forough Mortazavi

Academic Editor

PLOS ONE

1 February 2022

Dear Dr. Mortazavi,

PONE-D-21-28962

The relationship between happiness and self-rated health: A population-based study of 19499 Iranian adults

PLOS ONE

Thank you for your e-mail and the comments. We have revised the manuscript and now pleased to submit the second revision for your consideration. Hope you find revisions satisfactory.

Wish you all the best.

Yours sincerely

Ali Montazeri

Editor Comments:

Please kindly consider the points raised by reviewer 1 regarding item 28 and auto-correlation.

Thank you for your comment. The point was responded. Please see reviwer1.

In presenting the results of statistical analyses, please describe how the authors divided the sample into two groups of low and high happiness for logistic regression analyses. This section is in need of further clarification.

The following sentence was added to the statistical analysis as requested: 

Logistic regression analyses were performed to assess the relationship between happiness and independent variables, including participants’ health status. As such happiness as dependent variables was categorized into: ‘happy’ (scores ranging from 4 to 6) and ‘unhappy’ (scores ranging from 1 to 3).

Also, in the titles of tables 2 and 3, PLS replace 'happiness' with the correct description, i.e. 'Oxford happiness scores’.

Thank you. Done.

Reviewers'Comments to the Author

Reviewer #1 Prof. Ruut Veenhoven

My objection about auto-correlation is not met. The answer that the concerned questions on health were formulated differently does not convince. Without re-analysis of the data I cannot recommend acceptance.

Thank you for your comment. The correlation between happiness without item 28 and self-reported health examined and now was added to Appendix. 

Perhaps in future studies, if we are going to measure happiness in the sense of life satisfaction, there is a need to use an appropriate measure. In addition, item 28 of the Oxford Happiness Questionnaire is about self-rated health, and thus it might cause autocorrelation with the self-rated health measure, although the item 28 and the measure of self-rated health are worded differently. The latter is negative (I do not feel particularly healthy) while the former is positive and askes people to rate their current health. However, we did examine this correlation and found the correlation coefficient to be 0.17 (See S1 Appendix).

Reviewer #2: Dr. Vahid Rashedi

I read the manuscript with great interest and think the data reported in this study is valuable, and the authors have made all the corrections to the article entitled: "The relationship between happiness and self-rated health: A population-based study of 19499 Iranian adults".

Thank you for comments and time spent to review this paper.

---

## [Editor Report · Decision Letter 2]

21 Feb 2022

PONE-D-21-28962R2The relationship between happiness and self-rated health: A population-based study of 19499 Iranian adultsPLOS ONE

Dear Dr. Montazeri,

Thank you for submitting your manuscript to PLOS ONE. After careful consideration, we feel that it has merit but does not fully meet PLOS ONE’s publication criteria as it currently stands. Therefore, we invite you to submit a revised version of the manuscript that addresses the points raised during the review process.

We look forward to receiving your revised manuscript.

Kind regards,

Forough Mortazavi

Academic Editor

PLOS ONE

Additional Editor Comments:

Dear authors,

Thank you for revising your manuscript. Based on comments by reviewer #1 and my own evaluation, an important point raised still remains unaddressed. PLS take note that without the approval of reviewer # 1, the manuscript cannot be accepted for publication. I strongly recommend that you reanalyze the data after removing the items pointed out by reviewer #1.
---

## [Author Response · Author response to Decision Letter 2]

27 Feb 2022

Forough Mortazavi

Academic Editor

PLOS ONE

Thank you for your e-mail and the comments. We have revised the manuscript and now pleased to submit the third revision for your consideration. Hope you find revisions satisfactory.

Wish you all the best.

Yours sincerely

Ali Montazeri

Editor Comments:

Based on comments by reviewer #1 and my own evaluation, an important point raised still remains unaddressed. PLS take note that without the approval of reviewer # 1, the manuscript cannot be accepted for publication. I strongly recommend that you reanalyze the data after removing the item pointed out by reviewer #1.

Thank you. All comments were attended as suggested. The following revisions were applied:

1. Methods (Statistical analysis):

Data were explored using descriptive statistics, including frequency, percentage, mean and standard deviation. Logistic regression analyses were performed to assess the relationship between happiness and independent variables, including participants’ health status. However, since some eminent scholars [36] believe that there is an auto-correlation between item 28 and the self-rated health, we did reanalyze the data while item 28 (I do not feel particularly healthy) was excluded from the Oxford happiness score. As such for both with and without item 28 of the Oxford questionnaire, happiness as dependent variables were categorized into: ‘happy’ (scores ranging from 4 to 6) and ‘unhappy’ (scores ranging from 1 to 3). The results expressed as odds ratio and 95% confidence intervals. A significant level was set at P< 0.05.

2. Results:

The results are shown in Table 3. In addition, the results obtained from the same analysis when items 28 was excluded are shown in Table 4. The results almost were very similar and no significant difference was observed from the previous analysis except for age 18-24 (OR:1.191, 95% CI, P= 1.038-1.367, p= 0.013) and 6-9 years of education (OR:1.118, 95% CI, P= 1.024-1.221, p= 0.013).

3. Strengths and limitations

In addition, item 28 of the Oxford Happiness Questionnaire is about self-rated health, and thus it might cause autocorrelation with the self-rated health measure, although the item 28 and the measure of self-rated health are worded differently. The latter is negative (I do not feel particularly healthy) while the former is positive and askes people to rate their current health. However, as indicated in reanalysis of the data (Table 4), the findings did not show any major differences to our earlier analysis as shown in Table 3.

---

## [Decision Letter · Decision Letter 3]

7 Mar 2022

PONE-D-21-28962R3The relationship between happiness and self-rated health: A population-based study of 19499 Iranian adultsPLOS ONE

Dear Dr. Montazeri,

Thank you for submitting your manuscript to PLOS ONE. After careful consideration, we feel that it has merit but does not fully meet PLOS ONE’s publication criteria as it currently stands. Therefore, we invite you to submit a revised version of the manuscript that addresses the points raised during the review process. PLS cover the reviewer’s comments in three stages of review in detail.

We look forward to receiving your revised manuscript.

Kind regards,

Forough Mortazavi

Academic Editor

PLOS ONE

Reviewers' comments:

Reviewer's Responses to Questions

**Comments to the Author**

1. If the authors have adequately addressed your comments raised in a previous round of review and you feel that this manuscript is now acceptable for publication, you may indicate that here to bypass the “Comments to the Author” section, enter your conflict of interest statement in the “Confidential to Editor” section, and submit your "Accept" recommendation.

Reviewer #1: (No Response)

2. Is the manuscript technically sound, and do the data support the conclusions?

Reviewer #1: Partly

3. Has the statistical analysis been performed appropriately and rigorously? 

Reviewer #1: Yes

4. Have the authors made all data underlying the findings in their manuscript fully available?

Reviewer #1: (No Response)

5. Is the manuscript presented in an intelligible fashion and written in standard English?

Reviewer #1: Yes

6. Review Comments to the Author

Reviewer #1: In an earlier comment I noted that the the Oxford happiness scale involves an item on health, which causes auto correlation. I advised to recalculate, leaving this item out

I also noted that this questionaire covers broader matters than happiness as discussed in the introduction, and advised to calculate correlations for the few items on life-satisfaction separately

The authors did not recalculate, but sufficed mentioning these points under limitations. In my view, that is not acceptable

Still another point: the authors write Despite the importance of studying the relationship between happiness and health, only a few

81 small-scale studies have been conducted in Iran [21- 24] They are apparently unaware of the 337 findings on this topic listed in the World Database of happiness at https://worlddatabaseofhappiness.eur.nl/search-the-database/correlational-findings/#id=N1iFWn8BRfHHvZHVJh7X

7. PLOS authors have the option to publish the peer review history of their article (what does this mean?). If published, this will include your full peer review and any attached files.

Reviewer #1: **Yes: **Ruut Veenhoven

---

## [Author Response · Author response to Decision Letter 3]

8 Mar 2022

Dr. Forough Mortazavi

Academic Editor

PONE-D-21-28962

The relationship between happiness and self-rated health: A population-based study of 19499 Iranian adults

PLOS ONE

Thank you for your e-mail and the comments. We have revised the manuscript and now pleased to submit the fourth revision for your consideration. Hope you find revisions satisfactory.

Wish you all the best.

Yours sincerely

Ali Montazeri

---

## [Decision Letter · Decision Letter 4]

10 Mar 2022

The relationship between happiness and self-rated health: A population-based study of 19499 Iranian adults

PONE-D-21-28962R4

Dear Dr. Montazeri,

We’re pleased to inform you that your manuscript has been judged scientifically suitable for publication and will be formally accepted for publication once it meets all outstanding technical requirements.

Kind regards,

Forough Mortazavi

Academic Editor

PLOS ONE

Additional Editor Comments (optional):

Reviewers' comments:

Reviewer's Responses to Questions

**Comments to the Author**

1. If the authors have adequately addressed your comments raised in a previous round of review and you feel that this manuscript is now acceptable for publication, you may indicate that here to bypass the “Comments to the Author” section, enter your conflict of interest statement in the “Confidential to Editor” section, and submit your "Accept" recommendation.

Reviewer #1: All comments have been addressed

2. Is the manuscript technically sound, and do the data support the conclusions?

Reviewer #1: Yes

3. Has the statistical analysis been performed appropriately and rigorously? 

Reviewer #1: Yes

4. Have the authors made all data underlying the findings in their manuscript fully available?

Reviewer #1: Yes

5. Is the manuscript presented in an intelligible fashion and written in standard English?

Reviewer #1: Yes

6. Review Comments to the Author

Reviewer #1: (No Response)

7. PLOS authors have the option to publish the peer review history of their article (what does this mean?). If published, this will include your full peer review and any attached files.

Reviewer #1: **Yes: **Ruut Veenhoven

---

## [Editor Report · Acceptance letter]

15 Mar 2022

PONE-D-21-28962R4 

The relationship between happiness and self-rated health: A population-based study of 19499 Iranian adults 

Dear Dr. Montazeri:

I'm pleased to inform you that your manuscript has been deemed suitable for publication in PLOS ONE. Congratulations! Your manuscript is now with our production department. 

Kind regards, 

on behalf of

Dr. Forough Mortazavi 

Academic Editor

PLOS ONE